# Analytical Studies of Antimicrobial Peptides as Diagnostic Biomarkers for the Detection of Bacterial and Viral Pneumonia

**DOI:** 10.3390/bioengineering9070305

**Published:** 2022-07-11

**Authors:** Olalekan Olanrewaju Bakare, Arun Gokul, Marshall Keyster

**Affiliations:** 1Environmental Biotechnology Laboratory (EBL), Department of Biotechnology, University of the Western Cape, Cape Town 7535, South Africa; agokul5@gmail.com (A.G.); mkeyster@uwc.ac.za (M.K.); 2Department of Biochemistry, Faculty of Basic Medical Sciences, Olabisi Onabanjo University, Sagamu 120107, Ogun State, Nigeria; 3Department of Plant Sciences, Qwaqwa Campus, University of the Free State, Phuthadithjaba 9866, South Africa

**Keywords:** antimicrobial peptides, bacteria, viruses, databases, machine learning tool, diagnostics, receptors

## Abstract

Pneumonia remains one of the leading causes of infectious mortality and significant economic losses among our growing population. The lack of specific biomarkers for correct and timely diagnosis to detect patients’ status is a bane towards initiating a proper treatment plan for the disease; thus, current biomarkers cannot distinguish between pneumonia and other associated conditions such as atherosclerotic plaques and human immunodeficiency virus (HIV). Antimicrobial peptides (AMPs) are potential candidates for detecting numerous illnesses due to their compensatory roles as theranostic molecules. This research sought to generate specific data for parental AMPs to identify viral and bacterial pneumonia pathogens using in silico technology. The parental antimicrobial peptides (AMPs) used in this work were AMPs discovered in our previous in silico analyses using the HMMER algorithm, which were used to generate derivative (mutated) AMPs that would bind with greater affinity, in order to detect the bacterial and viral receptors using an in silico site-directed mutagenesis approach. These AMPs’ 3D structures were subsequently predicted and docked against receptor proteins. The result shows putative AMPs with the potential capacity to detect pneumonia caused by these pathogens through their binding precision with high sensitivity, accuracy, and specificity for possible use in point-of-care diagnosis. These peptides’ tendency to detect receptor proteins of viral and bacterial pneumonia with precision justifies their use for differential diagnostics, in an attempt to reduce the problems of indiscriminate overuse, toxicity due to the wrong prescription, bacterial resistance, and the scarcity and high cost of existing pneumonia antibiotics.

## 1. Introduction

Pneumonia is challenging because of its significant impact on global health [1]. It is characterized by specific infections with a distinct clinical source, presentation, epidemiology, and pathogenesis [2]. The disease arises from an infection in the lower respiratory tract, starting with inflammation and impaired tissue functioning [3]. Diagnosis of the disease may be difficult for an established respiratory condition, cardiac failure, and non-pneumonic respiratory infection because there is no evidence-based consensus on its clinical signs and predictive symptoms [4]. The diagnostic biomarkers used at present are produced in response to other conditions such as HIV and atherosclerosis [5]. The use of antibodies has remained questionable despite its high sensitivity due to challenges such as cross-reactivity [6]. The discovery of more specific and sensitive biomarkers for detecting pneumonia in patients’ samples is an ultimate goal of researchers to reduce the high mortality arising from the disease.

Antimicrobial peptides (AMPs) are cationic and amphiphilic molecules that act as host defense peptides [7]. AMPs are essential against drug resistance that causes serious health problems globally [8]. AMPs use diverse mechanisms to bind and interact with cell membranes to change the electrochemical potential, induce membrane damage, or penetrate large biomolecules such as proteins to destroy the cell morphology and membrane and cause cell death [8]. They have broad-spectrum antimicrobial activities against bacteria, viruses, fungi, cancers, and drug resistance [8] and possess immunomodulatory functions using diverse mechanisms such as structural changes, amino acid substitution, terminal acylation and amidation, and conjugation with cell penetration peptides [9]. AMPs can be engineered using site-directed mutagenesis because the natural ones derived from various organisms are not stable due to their short half-lives and severe proteolytic activities [10]. An AMP with desirable features would overcome these previously mentioned constraints for applications in biotechnology and medical sciences as potential diagnostic biomarkers.

Site-directed mutagenesis (SDM) is a molecular biology and bioinformatics method used to make specific and intentional changes to the DNA sequence and gene products [11]. It investigates the structure and biological activity of DNA, RNA, and protein molecules and is used in protein engineering. The application of site-directed mutagenesis to studying protein function has been widely illustrated. For instance, it has become an essential tool in biotechnology for the design of non-immunogenic antibodies for human therapeutics, which underscores the practical benefits of mutagenesis and protein engineering; it is a powerful technique in the study of protein function which allows the assessment of the particular amino acid side chains in a protein; and it is most commonly used in the study of enzymes and is also very useful in identifying key residues in protein–protein interactions [12]. Site-directed mutagenesis is often beneficial in generating AMP derivatives since amino acids of AMPs play crucial roles in their functioning [13]. Through SDM, the functioning of AMPs can be augmented, for instance, by increasing the interaction between the AMP and its target, thus increasing the action of the AMP. However, site-directed mutagenesis is still an underexplored tool to determine the function of a gene or improve a protein of interest because the principles guiding the use require a great deal of expertise [14].

In our previous studies, 19 antibacterial and 27 antiviral novel AMPs were identified using an in silico mathematical algorithm, HMMER, a machine learning tool whose codes are written in Figure 1, to identify three bacterial [15] and viral pathogens [16], respectively, in which the binding of these AMPs to the pneumonia pathogen receptors identified was carried out. Determining the amino acid residues within the binding interface that contribute the most to the binding affinity, known as hotspot residues, is imperative. Changes in these amino acids would significantly alter the structure and functioning of the identified putative AMPs. Thus, non-hotspot residues will then be mutated to increase the efficacy of binding. Using a similar approach, Williams et al. [17] demonstrated increased binding affinity of anti-HIV AMPs against p24, increasing the sensitivity and specificity of a lateral flow device to detect HIV 1 and 2 using AMPs instead of antibodies.

The present research aimed to use the parental anti-pneumonia AMPs generated using the HMMER machine learning tool from our previous studies demonstrated using in silico binding studies as templates to create derivative AMPs with potential greater affinity for the receptors. The site-directed mutagenesis technique was used to increase the binding efficacy of these potential anti-pneumonia AMPs. Increasing the binding affinity between the ligand and the receptor could improve the possible test specificity during molecular validation. A test kit’s ability to detect a pathogen at very low concentrations with high specificity and sensitivity is paramount in creating a perfect test kit to differentiate one pathogen from another and then bind to it with high affinity.

## 2. Materials and Methods

### 2.1. Data Retrieval

Nineteen putative antibacterial AMPs against *Streptococcus pneumoniae* (7), *Klebsiella pneumonia* (6), and *Acinetobacter baumanii* (5) [15] and twenty-seven putative antiviral pneumonia AMPs against respiratory syncytial virus (13) and the influenza A (8) and B (6) viruses [16] were selected and utilized for the identification of the mutation-sensitive amino acid residues.

### 2.2. Identification of Hotspot Residues

Before the commencement of in silico site-directed mutagenesis, essential amino acids have to be identified. The Knowledge-based FADE and Contacts (KFC) Server ((accessed on 2 February 2019) was employed for the precise identification of hotspot residues or mutation-sensitive residues to perform in silico site-directed mutagenesis effectively (Figure 2). The KFC Server uses two predictive models, shape specificity features and biochemical contacts, to display the hotspots’ predictive accuracy in identifying mutation-sensitive amino acid residues [18].

### 2.3. In Silico Site-Directed Mutagenesis (SDM)

An in silico SDM approach was used to generate derivative AMPs that displayed increased predicted binding affinity to the receptor proteins of the bacterial and viral pneumonia by substituting amino acid residues at regions distinct from the peptide hotspots using the parental AMPs against viral and bacterial pneumonia as templates. All amino acids were substituted to generate derivate AMPs and to maintain the predicted activity and functioning of the AMPs.

### 2.4. Physicochemical Properties

Submission of all derivative AMP sequences was carried out using the AMP characterization software BACTIBASE (physiochemical properties tab) (http://bactibase.pfba-lab-tun.org/physicochem (accessed on 1 June 2019)) [19] and the Antimicrobial Peptide Database (APD3) ((accessed on 1 July 2019)) [20] to determine the characteristics of each derivative AMP.

### 2.5. Prediction of Receptors

The receptor proteins from bacterial and viral pneumonia as identified in our previous studies were outer membrane proteins of *Acinetobacter baumannii* (**SCY38300.1**) and *Klebsiella pneumoniae* (**AIT02889.1**), pneumolysin of *Streptococcus pneumoniae* (**AJS15225.1**), nucleoproteins of influenza A (**pdb|3ZDP|C**) and B (**AAB72046.1**), and chain A matrix protein (**pdb|4D4T|A**) of respiratory syncytial virus.

### 2.6. In Silico 3D Structure Prediction

Prediction of the derivative AMPs’ 3D structures was carried out using the online software I-TASSER (Iterative Threading Assembly Refinement) (http://zhanglab.ccmb.med.umich.edu/I-TASSER (accessed on 1 August 2019)) [21]. The 3D structures of the AMPs were visualized using PyMol version 1.3 [22].

### 2.7. In Silico Protein–Protein Interaction Study

The docking interaction analysis of the antibacterial and antiviral pneumonia AMPs was carried out against their respective protein receptors, as identified in our previous study [15]. The online protein–protein interaction server PATCHDOCK was employed for this analysis [23]. Docking of each derivative AMP 3D structure was carried out against its specific protein receptor’s 3D structure, setting the RMSD as 4.0. Visualization of the 3D structure was carried out using PyMol version 1.3 (Figure 2) [22].

## 3. Results

### 3.1. Knowledge-Based FADE and Contacts (KFC) Analysis

The Knowledge-based FADE and Contacts (KFC) Server was used to identify hotspots or the subset of residues that determined the receptor proteins’ interface binding free energy. The results are tabulated in Table 1, showing the hotspot residues for each interaction between a specific ligand and its receptor. Any amino acid hotspot residues identified within the interaction site of the parental AMPs cannot be substituted during SDM analysis to maintain their functionality.

Table 1 above represents the amino acid residues of the bacterial and viral receptors used against the respective putative AMPs (parental AMPs) at the hotspot positions which play critical roles in the binding process and must not be substituted as they may change the functionality of the AMPs. BOPAM is the given name of the AMPs, with the following two letters being derivatives from the pathogens, for example, *Klebsiella pneumoniae* = KP.

### 3.2. In Silico Site-Directed Mutagenesis of the AMPs

After identifying the hotspot residues at the interaction sites of the anti-pneumonia AMPs and their receptors, site-directed mutagenesis was performed with these parental anti-pneumonia AMPs as input (Table 2). All amino acids substituted to generate derivate AMPs had similar characteristics to the amino acids present in the parental AMPs to maintain the predicted activity and functioning of the AMPs. BOPAM-AB1, 2, and 3 were mutated at position 10, where lysine was changed for arginine, while BOPAM-AB4 and 5 had leucine and isoleucine substituted for methionine at position 16. Moreover, BOPAM-SP1, 2, 3, 4, and 5 had serine or asparagine substituted for aspartate at position 25, while lysine was substituted for arginine in BOPAM-SP6 and 7 at position 36. Cysteine and tryptophan of BOPAM-KP1, 2, 3, and 7 were mutated at position 2, while glycine and serine of BOPAM-KP4, 5, and 6 were mutated at position 9, all with asparagine.

The antiviral AMPs were also subjected to site-directed mutagenesis, where lysine of BOPAM-INFA1, 2, 4, 6, and 7 at position 13 was substituted for arginine, while BOPAM-INFA3, 5, and 8 had glycine at position 7, which was substituted for threonine. BOPAM-INFB1 and 2 had histidine at position 5, which was substituted for arginine, while BOPAM-INFB3 had cysteine at position 12, which was substituted for serine. BOPAM-INFB 5, 6, and 4 had leucine at positions 8 and 15, which was substituted for methionine and phenylalanine, respectively. Lastly, BOPAM-RSV1, 2, 3, 4, 6, 7, and 8 had serine, threonine, valine, and tyrosine, which were mutated at position 3 with asparagine, while BOPAM-RSV5, 9, 10, 11, 12, and 13 had threonine and isoleucine which were substituted at position 13 with arginine.

### 3.3. Physicochemical Properties of the Derivative AMPs

Following the substitution of amino acids distinct from the hotspot residues, the mutated AMPs were again subjected to physicochemical analysis using APD3 to predict molecular masses, hydrophobicity, isoelectric points, and the Boman index, and BACTIBASE to predict the common amino acids, net charges, and half-lives. This analysis was carried out to ensure that the mutated AMPs retained the same characteristics as the parental ones. From the results observed in Table 3, all the derivative antibacterial pneumonia AMPs had improved performance in terms of the Boman index and hydrophobicity compared to the parental AMPs. However, BOPAM-KP1-3 and BOPAM-SP7 had slightly reduced hydrophobicity, with a higher Boman index. A similar result was generated for the site-directed mutagenesis of anti-HIV AMP8 to yield AMP8.1 used for HIV diagnosis with reduced hydrophobicity, but an improved binding potential to the HIV p24 protein [17]. An amino acid was substituted with one carrying a higher positive charge, and the overall charge of the AMP and the Boman index were increased. Therefore, it is expected that the derivative AMP would have a greater binding affinity to its receptor than its parental counterpart due to the specific amino acid substitution. However, the results indicate that BOPAM-SP3.1, 4.1, 5.1, and 6.1 had negative charges following SDM, while BOPAM-KP1.1 was neutral. With the discovery of negatively charged AMPs such as maximin H5, which contradicts the belief that AMPs with high antimicrobial activities are cationic, there is an indication that the charge might not affect an AMP’s activity [8].

From the results observed, all derivative antiviral pneumonia AMPs had improved performance in the Boman index and hydrophobicity compared to the parental AMPs. However, BOPAM-INFB3 and BOPAM-RSV3 and 13 had slightly reduced hydrophobicity, which does not interfere with an AMP’s binding potential as indicated above [8]. BOPAM-INFA5.1 and BOPAM-RSV2.1, 4.1, 6.1, 8.1, and 11.1 of the antiviral pneumonia AMPs had negative charges, with neutral charges observed for BOPAM-INFA1.1 and 4.1, BOPAM-INFB3.1 and 6.1, and BOPAM-RSV3.1 and 7.1, as indicated in Table 3 below.

### 3.4. Structure Prediction Using I-TASSER

Figure 3 below shows the structures of the derivative anti-pneumonia AMPs with extended-partial α-helix and β-hairpin structures following SDM. With the slight changes in the structural characteristics of the AMPs post-SDM, the derivative anti-pneumonia AMPs still contained structural features associated with this class of peptides.

Table 4 below shows the quality output from the I-TASSER server after predicting the derivative anti-pneumonia AMPs using the C-score, TM-score, and RMSD as measures. The C-scores of all derivative AMPs were within the range of −5 to 2, indicating accurate predictions from the templates used with the correct topology. The C-scores of the derivative AMPs did not deviate from those of the parental AMPs from which they were derived. The TM-scores were greater than 0.17, showing that the 3D structures of the derivative AMPs were not random predictions. The RMSD values were above 1 Å, further proving that the AMPs were correctly predicted.

### 3.5. Docking Interaction Analysis of the Derivative AMPs with Bacterial and Viral Receptors Using PATCHDOCK

Table 5 below shows the binding scores of the antibacterial and antiviral pneumonia AMPs from the PATCHDOCK server with their respective receptors identified in the previous study. All binding scores of the derivative AMPs were compared with the parental AMPs for selection as candidate anti-pneumonia AMPs to be used in a lateral flow device (LFD). BOPAM-AB1.1 and BOPAM-AB5.1 returned higher binding scores than the parental AMPs against the outer membrane receptor protein of Acinetobacter baumannii. Moreover, BOPAM-SP1.1, 3.1, 4.1, and 7.1 yielded improved binding scores against the pneumolysin of Streptococcus pneumoniae, while a similar result was observed for BOPAM-KP1.1, 3.1, 5.1, and 6.1 against the iron-regulated outer membrane protein of *Klebsiella pneumoniae*. 

All mutated anti-influenza A AMPs returned higher binding scores than the respective parental AMPs, where improved binding efficacies were recorded for the AMPs against the nucleoprotein of influenza A virus except for BOPAM-INFA5.1. Only BOPAM-INFB5.1 had a high binding score against the *influenza B* virus nucleoprotein, while BOPAM-RSV3.1, BOPAM-RSV6.1, BOPAM-RSV9.1, BOPAM-RSV10.1, and BOPAM-RSV13.1 all had improved binding scores against *respiratory syncytial* virus chain A.

All the derivative AMPs above (Table 4) had increased binding affinity to their receptors compared to their respective parental AMPs (Figure 4). In other words, they bound more tightly when compared to the separate parental AMPs at the same positions. The mutation due to amino acid substitution in these derivative AMPs did not change their 3D structures compared to those of the parental AMPs and displayed a better-predicted fit at the binding pockets of bacterial and viral pathogens’ receptors compared to the parental AMPs.

There was no change in the binding region or orientation of the derivative AMPs with their receptors after substituting the amino acid residues of the parental AMPs, as indicated in the previous studies. These peptides’ diagnostic tendency could be attributed to their small size, allowing them access to membrane receptors’ biological surfaces. The receptors used in this research were carefully identified with residues seated at the membrane’s extracellular surface for accessibility during binding.

The interaction studies indicate that the BOPAM-SPs bound pneumolysin at the extracellular site of *Streptococcus pneumoniae* at the N-terminal domain; BOPAM-KPs bound the extracellular outer membrane protein receptor of *Klebsiella pneumoniae* at the N-terminal domain; and BOPAM-ABs bound the extracellular C-terminal domain of the outer membrane protein receptors of *Acinetobacter baumannii*. In the same manner, the binding affinity of the antiviral AMPs occurred at the exposed regions of their receptors, with BOPAM-INFAs binding the nucleoprotein at the N-terminal domain of *influenza A* virus and BOPAM-INFBs binding the nucleoprotein of *influenza B* virus at N-terminal domain, while BOPAM-RSVs bound both chain A proteins of *respiratory syncytial* virus at the N-terminal domains.

## 4. Discussion

Various shortcomings have been suffered from using different diagnostic systems of bacteria and viruses. Antibodies, which are currently the gold standard, also suffer from limitations such as cross-reactivity and sequence coverage. Other shortcomings include overlapping epitopes between the detected and natural host antibodies, resulting in false negative results, complex sample preparation, time consumption, and non-specific binding to non-target molecules [24]. Antimicrobial peptides have shown tremendous potential in circumventing the drawbacks of these systems and appear to be the favorite choice in diagnostic development due to their numerous properties that overcome these shortcomings, such as their small size, ease of modification, high stability, and minor/non-toxicity. The HMMER machine learning tool was employed to construct predictive models to identify sensitive and specific AMPs against *Streptococcus pneumonia, Klebsiella pneumonia, Acinetobacter baumannii,* respiratory syncytial virus, and the influenza A and B viruses in our previous studies [15,16]. This research employed the predicted AMPs as parental peptides to develop derivative AMPs using site-directed mutagenesis.

The Knowledge-based FADE and Contacts (KFC) Server was used to identify the amino acid residue “hotspots” that contributed to the receptor proteins’ interface binding free energy to differentiate the mutation-sensitive residues. This means that these amino acid residues contribute the most to the free energy involved in the binding to their respective receptors. The use of the KFC Server offers two predictive models for accuracy dependent on the shape specificity features or the biochemical contacts [18]. After that, SDM analysis was performed to generate derivative AMPs to identify the ones that would lead to a higher predicted binding energy to the respective receptors of the pathogens [25]. The amino acid residues were substituted using SDM to maintain the predicted activity and functionality of the AMPs. This ensures that the enormous potential of the AMPs is realized to achieve the desired goal of sensitive identification of the pneumonia pathogens.

Physicochemical analysis of the derivative AMPs was conducted to ascertain their conformity to the known AMPs using criteria such as the charge, polarity, hydrophobicity, and Boman index. The reduced hydrophobic values of BOPAM-KP4.1, BOPAM-SP1.1, 2.1, and 7.1, BOPAM-INFA1.1, 3.1, 4.1, 5.1, 6.1, and 8.1, and BOPAM-RSV2.1, 7.1, 9.1, 10.1, and 13.1 are related to a reduced peptide helicity, poor self-associating ability in an aqueous phase, and reduced antimicrobial activity. Peptides with increased hydrophobicity can penetrate deeper into organisms, causing severe hemolysis by forming channels [26]. Thus, mutated antibacterial and antiviral AMPs with increased hydrophobicity could potentially penetrate the membrane core. More recently, AMPs from sugar-functionalized phosphonium polymers were reported to require the hydrophilic domains for their molecular structure to exert antibacterial activities against Gram-negative *Escherichia coli* and Gram-positive *Staphylococcus aureus* [27]. There is a positive correlation between cationic AMPs and antimicrobial properties (Table 3), which shows conformity with ideal AMPs [28]. Nonetheless, the net charge of mutated AMPs with negative charges does not interpret an absence of antimicrobial activities since some negatively charged AMPs have recently been discovered: for example, surfactant-associated anionic peptides in the APD3 database (AP00528) with a net charge of −5 which have antibacterial activity, and maximin H5 with a charge ranging between −1 and −7 which displays bacterial growth inhibition against *Listeria monocytogenes* [29]. The isoelectric point is a function of individual amino acid residues in the peptides, which relates to their solubility properties in acid or alkaline solutions regardless of their charges. A negative Boman index directly correlates with hydrophobic peptides with a high protein binding potential. Notwithstanding, the positive Boman index in some peptides has been accounted for with the capacity to distinguish HIV in a lateral flow device [30]. All the mutated peptides exhibited a short half-life because they are not stable, and half-life values as low as one have been reported for AMP molecules used for HIV diagnosis [31].

The structure prediction results utilizing the C-scores from I-TASSER relate to the modeling of a template’s certainty for predicting the structure quality, that is, the anticipated model’s distance and its local structures [32]. TM and RMSD scores estimate structural model closeness and accuracy in two structures when the local structure is known. The outcomes of the peptides’ structures demonstrated that they conformed to known AMPs. This outcome relates to the work of [15,16], where binding geometry scoring was utilized as the criteria for determining applicant AMPs for pneumonia diagnostics. These perceptions were additionally affirmed utilizing an in-house lateral flow device in which the putative AMPs were utilized to recognize HIV in patient samples [30].

The results from this research show that all the putative peptides displayed in Table 5 could be pursued for molecular validation as applicant specialists for the detection of bacterial and viral pneumonia pathogens. Furthermore, the docking interaction results display only the mutated peptides with increased binding potential and atomic contact energy to the respective receptors. As shown in this analysis, these parameters are significant for the determination of novel peptides for detecting diseases through the development of an LFD device [30]. This research is in line with Williams et al. [30], where parental anti-HIV antimicrobial peptides were mutated using SDM analysis to increase their binding potential against the HIV nucleoprotein, and subsequent molecular validation was carried out for the construction of an LFD device. Thus, the putative anti-pneumonia AMPs discovered in this research have better binding energy than the parental AMPs. This offers promising perspectives for sensitive diagnosis and differentiation of bacterial from viral pneumonia in patients’ blood samples. It also allows health practitioners to forecast timely and correct treatment regimens for patients with this condition.

## 5. Conclusions

This research work used parental anti-pneumonia AMPs as templates to identify putative derivative AMPs that can function in the potential differential diagnosis of pneumonia. The derivative AMPs have an improved binding potential to the pneumonia receptors, with greater affinity, demonstrated through in silico binding studies. Thus far, 10 antibacterial and 13 antiviral pneumonia AMPs with improved binding specificity and accuracy against pneumonia pathogens have been identified. These AMPs offer promising perspectives to compete with current commercially available pneumonia diagnostic biomarkers to mitigate their shortcomings.

## 6. Future Work

Future work will include the in vitro study of the anti-pneumonia activity of the mutated peptides. Moreover, the EC_50_ of all the AMPs and their diagnostic or selective index will be assessed for optimization. The anti-pneumonia activity of these derivative AMPs will be assessed on different pseudotypes of the pneumonia pathogens to determine their broad-spectrum activity. Finally, the binding complex formed between the pathogen receptors and derivative AMPs will be solved using structural biology to validate the in silico binding study’s observations.

## Figures and Tables

**Figure 1 bioengineering-09-00305-f001:**
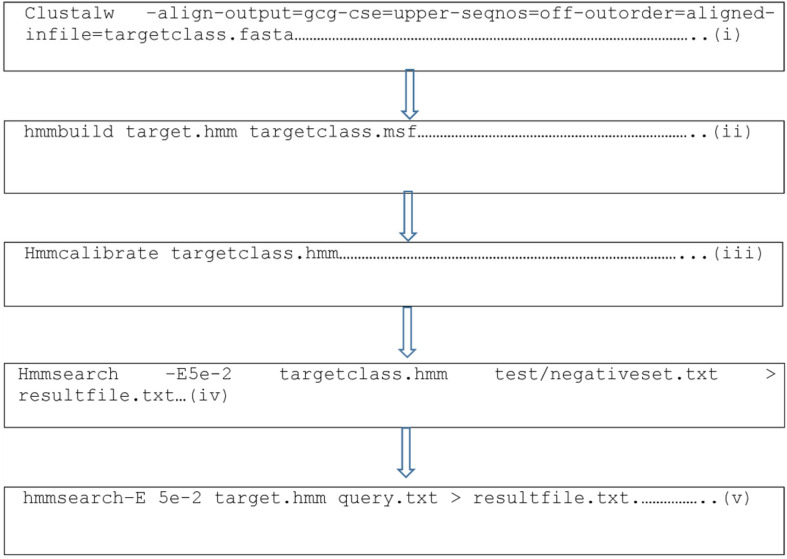
Representative codes for the generation of the parental putative antimicrobial peptides from the HMMER algorithm, a machine learning tool, where (**i**) indicates the generation of Clustal for the training datasets which yields the msf file; (**ii**) generates the hmm file from the msf file using hmmbuild; (**iii**) calibrates the hmm profile; (**iv**) searches for matching motifs in the test or negative datasets; and (**v**) searches for motifs in proteomes of other organisms for the discovery of new antimicrobial peptides.

**Figure 2 bioengineering-09-00305-f002:**
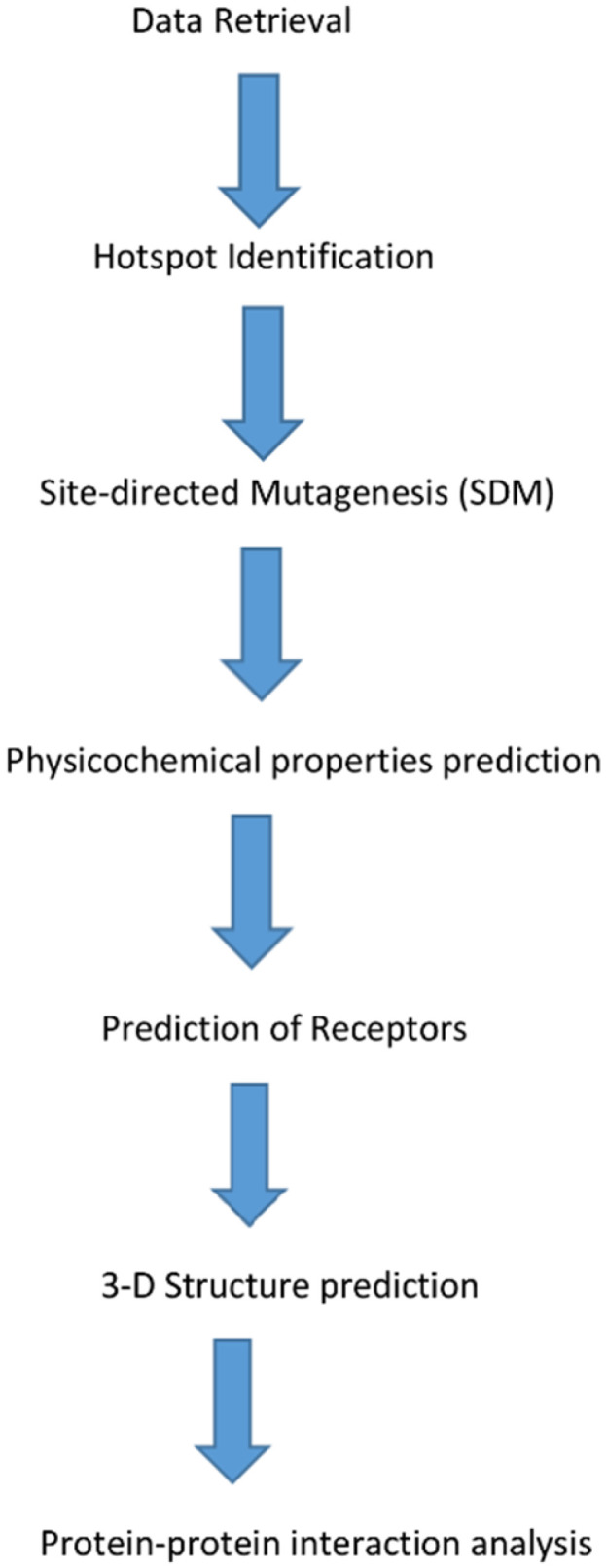
Schematic diagram of the processes involved in predicting derivative antimicrobial peptides (AMPs).

**Figure 3 bioengineering-09-00305-f003:**
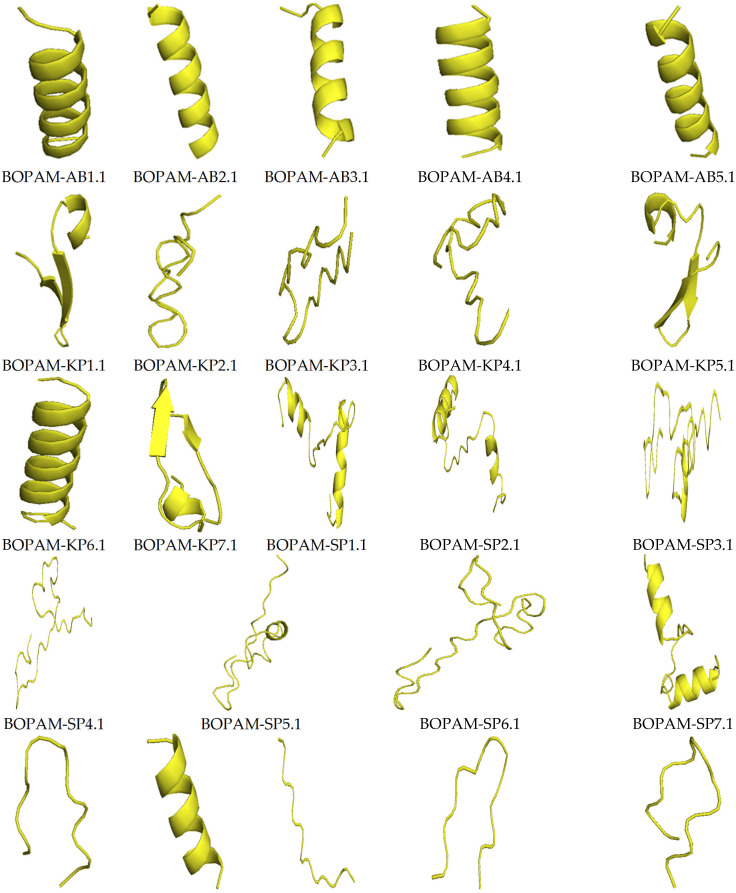
Structures of the derivative anti-pneumonia AMPs using I-TASSER. The AMPs showed different secondary structures including alpha-helix, beta-sheet, and extended conformations.

**Figure 4 bioengineering-09-00305-f004:**
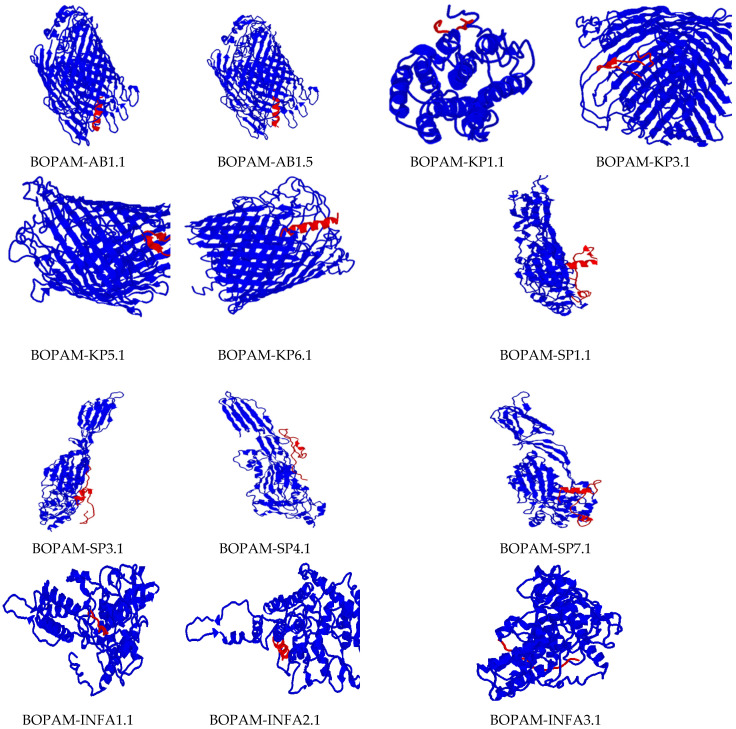
Docking interaction analysis of the mutated Ap-AMPs with their respective receptors using PATCHDOCH, visualized using PyMOL. The blue color represents the receptors, while the red color represents the AMPs.

**Table 1 bioengineering-09-00305-t001:** Identification of mutation-sensitive amino acid residue hotspots using KFC.

S/N	Interacting Amino Acids of the Receptors	Interacting Amino Acids of the Putative AMPs
1	*Acinetobacter baumannii* Outer Membrane Protein: Arg188A, Arg358A, Asn597A, Arg600A, Val612A, Asp644A, Gln646A, Asp739A, Gln787A, Leu789A, Asn792A, Asn794A	BOPAM-AB1: Phe1a, Leu2a, Val5a, Leu9a, Ser11a, Ser14a, Gly15a, Leu16a, Leu17a
2	*Acinetobacter baumannii* Outer Membrane Protein: Glu180A, Arg188A, Ser189A, Arg203A, Ser206A, Arg207A, Arg600A, Val645A, Gln646A, Tyr736A, Arg737A, Gln787A, Leu789A, Arg790A	BOPAM-AB2: Phe2a, Gly6a, Lys7a, Leu9a, Ser11a, Gly15a, Leu16a, Leu17a
3	*Acinetobacter baumannii* Outer Membrane Protein: Arg176A, Arg188A, Ser189A, Arg203A, Asp204A, Asp431A, Ile433A, Leu473A, Leu595A, Asn597A, Ala598A, Arg600A, Asp644A, Leu789A, Arg790A, Asn793A	BOPAM-AB3: Phe1a, Phe2a, Pro3a, Leu8a, Leu9a, Leu13a, Phe14a, Leu17a
4	*Acinetobacter baumannii* Outer Membrane Protein: Arg188A, Ser189A, Gln587A, Asn597A, Ala598A, Val612A, Asp644A, Gln646A, Phe648A, Tyr736A, Asp739A, Gln787A, Leu789A	BOPAM-AB4: Phe1a, Phe2a, Ile4a, Val5a, Lys7a, Leu8a, Leu9a, Lys10a, Ser14a
5	*Acinetobacter baumannii* Outer Membrane Protein: Thr184A, Arg188A, Ser189A, Arg600A, Val612A, Asp644A, Val645Aa, Gln646A, Phe648A, Tyr736A, Asp739A, Gln787A, Leu789A	BOPAM-AB5: Thr4a, Gly6a, Lys7a, Ala8a, Lys11a, Arg14a, Ala15a, Asn17a
6	Influenza A Virus Nucleoprotein: Ser165A, Leu166A, Arg267A, Gly268A, Val270A, His272A, Phe338A, Glu339A. Asp340A, Arg342A, Val343A, Pro453A, Ser457A, Leu479A, Tyr487A	BOPAM-INFA1: Pro2a, Phe4a, Ile5a, Asp6a, Gly7a, Gln8a, Val9a
7	Influenza A Virus Nucleoprotein: Ser165A, Leu264A, Ile265A, Arg267A, Asp340A, Arg342A, Val343A, Pro453A, Val456A, Ser457A, Phe458A, Pro477A, Leu479A, Tyr487A	BOPAM-INFA2: Pro2a, Val3a, Leu5a, Ile9a, Gln10a, Lys14a
8	Influenza A Virus Nucleoprotein: Glu339A, Arg342A, Val343A, Ala387A, Thr390A, Asn395A, Pro453A, Glu454A, Asp455A, Val456A, Ser457A, Phe458A, Gly462A, Val463A, Leu479A	BOPAM-INFA4: Phe4a, Ile5a, Asp6a, Gly7a, Gln8a, Val9a, Pro10a, Gln14a
9	Influenza A Virus Nucleoprotein: Leu166A, Arg267A, Val270A, His272A, Glu339A, Asp340A, Val343A, Ser457A, Phe458A, Val463A, Pro477A, Leu479A	BOPAM-INFA5: Thr1a, Thr3a, Phe4a, Ile5a, Val9a, Ile11a, Gln13a, Gln14a
10	Influenza A Virus Nucleoprotein: Ser165A, Arg267A, Glu339A, Asp340A, Val343A, Glu454A, Asp455A, Ser457A, Phe458A, Val463A, Pro477A, Leu479A	BOPAM-INFA6: Pro2a, Phe4a, Ile5a, Asp6a, Gly7a, Val9a, Pro12a
11	Influenza A Virus Nucleoprotein: Ser165A, Leu166A, Leu264A, Ile265A, Arg267A, Phe338A, Asp340A, Val343A, Asn395A, Pro453A, Asp455A, Val456A, Ser457A, Phe458A, Tyr487A	BOPAM-INFA7: Pro2a, Val3a, Ile4a, Asp6a, Gln10a, Val11a, Phe12a
12	Influenza A Virus Nucleoprotein: Ser165A, Ile265A, Arg267A, Gly268A, Val270A, His272A, Ala336A, Glu339A, Arg342A, Val343A, Ser344A, Ile347A, Ala387A, Arg389A, Pro453A, Phe458A	BOPAM-INFA8: Thr1a, Pro2a, Thr3a, Ile5a, Val5a, Glu13a, Gln14a
13	Influenza B Virus Nucleoprotein: Ser289A, Ala290A, Val323A, Val324A, Arg325A, Arg398A, Arg447A, Glu452A, Met503A, Ser507A, Gly514A	BOPAM-INFB1: Met1a, Val3a, Ser4a, Arg6a, Trp7a, Thr8a, Phe9a, Leu10a, Val12a
14	Influenza B Virus Nucleoprotein: Arg116A, Lys125A, Gly151A, Arg170A, Arg235A, Glu253A, Arg256A, Phe257A	BOPAM-INFB2: Ser4a, Arg6a, Thr8a, Phe9a, Met11a, Val12a, Pro13a, Pro14a
15	Influenza B Virus Nucleoprotein: Arg116A, Leu119A, Ala120A, Asp123A, Lys125A, Asp137A, Glu140A, Lys142A, Glu143A, Thr150A, Gly151A, Pro234A	BOPAM-INFB3: Leu1a, Asn2a, Pro5a, Gln6a, Leu7a, Leu8a, Leu10A,
16	Influenza B Virus Nucleoprotein: Arg116A, Leu119A, Ala120A, Asp123A, Lys125A, Asp137A, Glu140A, Thr150A, Gly152A, Thr153A, Pro234A, Pro422A, Ala423A	BOPAM-INFB4: Leu4a, Gln5a, Leu7a, Leu8a, Leu10a, Lys1a, Val12a, Pro13a, Leu15a
17	Influenza B Virus Nucleoprotein: Aeg116A Leu119A, Ala120A, Lys125A, Phe129A, Asp137A, Glu140A, Glu143A, Gly151A, Gly152A, Thr153A, His201A, Thr232A, Pro234A, His420A	BOPAM-INFB5: Thr2a, Ile3a, Leu4a, Leu6a, Leu7a, Leu10a, Lys11a, Gln14a, Leu15a
18	Influenza B Virus Nucleoprotein: Gln203A, Ala229A, Ile244A, Val322A, Val323A, Arg325A, Val328A, Ser330A, Tyr394A, Glu395A, Asp396A, Gly449A, Met508A, Gly514A, Ala516A	BOPAM-INFB6: Leu1a, Asn4a, Leu7a, Cys9a, Asn11a, Asn13a, Pro14a, Gln15a
19	Respiratory Syncytial Virus Chain A Protein: Trp37A, Cys93A, Asn95A, Val96A, Tyr199A, Ser200A, Ile225A, Asp227A, Ala230A	BOPAM-RSV1: Ile1a, Ile5a, Glu7a, Glu8a, Cys12a, Lys15a, Phe16a
20	Respiratory Syncytial Virus Chain A Protein: Trp37A, Ile92A, Cys93A, Asn95A, Val96A, Tyr199A, Ser200A, Leu203A, Ile225A, Asp227A	BOPAM-RSV2: Asn6a, Glu7a, Ile9a, Asp10a, Leu13a, Ser14a
21	Respiratory Syncytial Virus Chain A Protein: Trp37A, Met40A, Ile92A, Cys93A, Asn95A, Tyr199A, Ser200A, Leu203A, Ile225A, Asp227A	BOPAM-RSV3: Asn1a, Asp4a, Val5a, Lys8a, Ile9a, Asn12a, Thr13a
22	Respiratory Syncytial Virus Chain A Protein: Leu27A, Ile30A, Arg31A, Asp34A	BOPAM-RSV4: Val2a, Glu4a, Ile5a, Ala7a, Asn11a
23	Respiratory Syncytial Virus Chain A Protein: Trp37A, Ile92A, Cys93A, Leu128A, Thr130A, Tyr199A, Ser200A, Ile225A, Asp227A	BOPAM-RSV5: Ser6a, Ala7a, Gln8a, Asn10a, Lys11a, Asn15a
24	Respiratory Syncytial Virus Chain A Protein: Trp37A, Pro39A, Met40A, Cys93A, Thr130A, Tyr199A, Ser200A, Leu203A, Val226A, Asp227A	BOPAM-RSV6: Lys1a, Gln4a, Ile5a, Ala8a, Ile9a, Gln12a
25	Respiratory Syncytial Virus Chain A Protein: Trp37A, Cys93A, Asn95A, Thr130A, Tyr199A, Ser200A, Leu203A, Ile225A, Asp227A	BOPAM-RSV7: Lys6a, Asn10a, Gln11a, Ile13a, Asn14a
26	Respiratory Syncytial Virus Chain A Protein: Trp37A, Pro39A, Met40A, Cys93A, Lys125A, Thr130A, Tyr199A, Ser200A, Leu203A, Ile225A, Asp227A	BOPAM-RSV8: Ile2a, Asn4a, Phe5a, Ser9a, Leu12a, Leu13a, Ser14a
27	Respiratory Syncytial Virus Chain A Protein: Trp37A, Met40A, Cys93A, Asn95A, Lys125A, Leu128A, Thr130A, Tyr199A, Ser200A, Leu203A, Ile225A, Asp227A	BOPAM-RSV9: Ile2a, Ser3a, Lys4a, Thr6a, Asn7a, Asn10a, Thr11a
28	Respiratory Syncytial Virus Chain A Protein: Trp37A, Met40A, Cys93A, Asn95A, Thr130A, Tyr199A, Ser200A, Leu203A, Ile225A, Asp227A	BOPAM-RSV10: Ser6a, Asn10a, Thr11a, Thr14a, Asn15a, Ile16a
29	Respiratory Syncytial Virus Chain A Protein: Trp37A, Ser71A, Phe90A, Thr91A, Ile92A, Thr130A, Tyr199A, Ser200A, Leu203A, Ile225A, Val226A, Asp227A	BOPAM-RSV11: Asn1a, Val2a, Asp10a, Asn12a, Ala14a, Asp15a
30	Respiratory Syncytial Virus Chain A Protein: Glu1A, Lys27A, Asn56A, Asn58A, Arg79A, Pro111A, Cys112A, Glu113A, Ile114A	BOPAM-RSV12: Asn4a, Ile5a, Asn8a, Lys12a, Phe13a, Ile16a
31	Respiratory Syncytial Virus Chain A Protein: Trp37A, Tyr199A, Ser200A, Leu203A, Ile225A, Asp227A, Ala230A	BOPAM-RSV13: Leu5a, Glu7a, Lys8a, Asp11a, Arg12a
32	*Streptococcus pneumoniae* Pneumolysin: Thr57A, Ser58A, Asp59A, Met97A, Thr98A, Tyr99A, Ser100A, Lys196A, Ile198A, Thr201A, Ser203A, Asp205A, Ala206A, Asp212A, Ser239A, Ala241A, Ser330A, Thr332A, Phe335A, Val341A, Thr343A	BOPAM-SP1: Arg3a, Asp4a, Asp5a, Arg6a, Cys8a, Met12a, Ile24a, Thr26a, Phe27a, Ser34a, Ile38a, Cys39a, Asn43a, Gly44a
33	*Streptococcus pneumoniae* Pneumolysin: Ser58A, Asp59A, Thr98A, Tyr99A, Ser100A, Ile101A, Gln149A, Glu151A, Ile198A, Thr201A, Ser203A, Asp205A, Ser239A, Ala241A, Thr332A, Phe335A, Val341A, Thr343A	BOPAM-SP2: Arg3a, Asn4a, Cys8a, Met12a, Thr24a, Thr26a, Phe27a, His29a, Ser34a, Ile38a, Asn41a, Lys42a
34	*Streptococcus pneumoniae* Pneumolysin: Glu42A, Ser254A, Ser256A, Glu277A, Gln280A, Ile281A, Asn284A, Thr356A, Ala357A, Thr358A, Arg359A, Leu447A, Val448A, Asn470A, Asp471A	BOPAM-SP3: Phe1a, His13a, His14a, Gln15a, Lys16a, Leu17a, Val18a, Phe19a, Asp23a, Asn28a, Cys32a, Ala33a, Ile35a, Leu37a, Met38a
35	*Streptococcus pneumoniae* Pneumolysin: Val45A, Glu47A, Ser254A, Ser256A, Glu277A, Gln280A, Ile281A, Lys354A, Thr356A, Tyr358A, Arg359A, Arg419A, Pro446A, Leu447A, Val448A, Arg449A, Val468A, Asn470A, Asp471A	BOPAM-SP4: Arg2a, His3a, His13a, His14a, Gln15a, Lys16a, Leu17a, Val18a, Phe20a, Asp23a, Ser27a, Asn28a, Lys32a, Met38a, Ile44a
36	*Streptococcus pneumoniae* Pneumolysin: Lys19A, Leu20A, His23A, Glu26A, Val78A, Asp79A, Glu80A, Leu83A, Glu84A, Glu159A, Lys162A, Ser167A, Glu170A, Glu231A, Val351A	BOPAM-SP5: Phe1a, Arg2a, His3a, Glu4a, Phe20a, Val24a, Lys29a, Gly30a, Ile34a, Ile35a, Gly36a, Met38a
37	*Streptococcus pneumoniae* Pneumolysin: Glu42A, Thr253A, Ser254A, Lys255A, Ser256A, Glu277A, Ile281A, Asn284A, Ala357A, Tyr358A, Arg359A, Pro446A, Leu447A, Val448A, Asn470A	BOPAM-SP6: Glu11a, His14a, Gln15a, Lys16a, Leu17a, Val18a, Phe31a, Leu37a, Met38a
38	*Streptococcus pneumoniae* Pneumolysin: Thr55A, Thr57A, Asp59A, Met148A, Gln149A, Tyr150A, Glu151A, Lys164A, Phe165A, Asn194A, Lys196A, Glu264A, Lys268A, Val270A, Val272A, Ile314A, Glu315A, Phe344A	BOPAM-SP7: Arg3a, Cys8a, Asn25a, Phe27a, Asp34a, Ile38a, Lys40a, Asp41a, Lys42a, Asn43a, Gly44a
39	*Klebsiella pneumoniae* Iron-Regulated Outer Membrane Protein: Pro3A, His4A, Glu11A, Phe34A, Ser73A, Leu113A, Val115A, Phe142A, Arg225A, Asp227A, Glu228A, Tyr229A, Glu258A, Phe301A	BOPAM-KP1: Lys3a, Glu4a, Glu5a, Gly6a, Ser8a, Ser9a, His11a, Cys12a, Ser13a, Pro14a, Trp19a, Glu21a
40	*Klebsiella pneumoniae* Iron-Regulated Outer Membrane Protein: Ala94A, Arg108A, Thr109A, Ser111A, Arg112A, Tyr256A, Phe319A, Pro321A, Pro323A, Ser332A, Ser334A, Phe376A, Tyr501A, Tyr507A, Ser519A, Arg578A, Val620A, Thr621A, Gln696A	BOPAM-KP2: Lys3a, Tyr4a, Val5a, Lys7a, Gly9a, Leu10a, Asn11a, Gly13a, Gln15a, Lys17a, Ile18a, Asp19a, Asn20a
41	*Klebsiella pneumoniae* Iron-Regulated Outer Membrane Protein: Arg93A, Lys500A, Tyr501A, Leu515A, Leu517A, Thr561A, Ile562A, Val563A, Val564A, Asp576A, Val609A, Trp618A, Val620A, Thr621A, Ala654A, Ala655A, Arg697A, Pro704A	BOPAM-KP3: Met1a, Lys3a, His4a, Val6a, Lys7a, Leu8a, Val14a, Gln15a, Cys16a
42	*Klebsiella pneumoniae* Iron-Regulated Outer Membrane Protein: Gln81A, Asn85A, Thr109A, Tyr256A, Phe319A, Pro321A, Leu325A, Ser332A, Ser334A, Ser336A, Gln338A, Gln374A, Tyr507A, Ile518A, Ile693A, Gln696A, Arg697A, Ala698A, Leu700A, Leu711A	BOPAM-KP4: Ile1a, His2a, His3a, Glu4a, Ala5a, Lys7a, Gly8a, Tyr10a, Pro12a, Tyr13a, Leu14a, Trp16a, Leu18a
43	*Klebsiella pneumoniae* Iron-Regulated Outer Membrane Protein: Ala22A, Gln23A, Gly52A, Gln53A, Glu56A, Glu123A, Lys149A, Asp158A, Glu160A, Phe590A, Asp595A, Ala635A, Ala672A, Lys677A, Thr728A	BOPAM-KP5: Arg7a, Cys12a, Ser13a, Ala14a, Ser15a, Leu16a, Lys17a, Cys18a, Trp19a, Phe20a
44	*Klebsiella pneumoniae* Iron-Regulated Outer Membrane Protein: Ser79A, Gln81A, Asn85A, Met88A, Ala94A, Arg108A, Thr109A, Tyr256A, Phe376A, Tyr507A, Ile518A, Ser519A, Ile693A, Gln696A, Ala698A, Leu700A	BOPAM-KP6: Ala5a, Lys6a, Ala8a, Glu11a, Lys13a, Cys15a, Lys16a, Leu18a, Ala19a, Lys20a, Lys21a
45	*Klebsiella pneumoniae* Iron-Regulated Outer Membrane Protein: Ser79A, Gln81A, Asn85A, Gly87A, Met88A, Arg93A, Ala94A, Arg108A, Thr109A, Ser111A, Arg112A, Tyr256A, Ser234A, Lys500A, Tyr501A, Tyr502A, Tyr507A, Ile518A, Ile562A, Ile693A, Gln696A, Arg697A, Tyr712A	BOPAM-KP7: Leu1a, Arg3a, Glu4a, Val8a, Ser9a, His11a, Cys12a, Leu16a, Cys18a, Arg20a, Met22a

**Table 2 bioengineering-09-00305-t002:** Site-directed mutagenesis of the anti-pneumonia AMPs.

S/N	Parental AMP	Amino Acid and Position	Mutated AMP	Amino Acid and Position
1	BOPAM-AB1	K-10	BOPAM-AB1.1	R-10
2	BOPAM-AB2	K-10	BOPAM-AB2.1	R-10
3	BOPAM-AB3	K-10	BOPAM-AB3.1	R-10
4	BOPAM-AB4	L-16	BOPAM-AB4.1	M-16
5	BOPAM-AB5	I-16	BOPAM-AB5.1	M-16
6	BOPAM-KP1	C-2	BOPAM-KP1.1	N-2
7	BOPAM-KP2	W-2	BOPAM-KP2.1	N-2
8	BOPAM-KP3	W-2	BOPAM-KP3.1	N-2
9	BOPAM-KP4	G-9	BOPAM-KP4.1	N-9
10	BOPAM-KP5	S-9	BOPAM-KP5.1	N-9
11	BOPAM-KP6	G-9	BOPAM-KP6.1	N-9
12	BOPAM-KP7	C-2	BOPAM-KP7.1	N-2
13	BOPAM-SP1	N-25	BOPAM-SP1.1	D-25
14	BOPAM-SP2	N-25	BOPAM-SP2.1	D-25
15	BOPAM-SP3	S-25	BOPAM-SP3.1	D-25
16	BOPAM-SP4	S-25	BOPAM-SP4.1	D-25
17	BOPAM-SP5	S-25	BOPAM-SP5.1	D-25
18	BOPAM-SP6	L-36	BOPAM-SP6.1	R-36
19	BOPAM-SP7	L-36	BOPAM-SP7.1	R-36
20	BOPAM-INFA1	L-13	BOPAM-INFA1.1	R-13
21	BOPAM-INFA2	L-13	BOPAM-INFA2.1	R-13
22	BOPAM-INFA3	G-3	BOPAM-INFA3.1	T-3
23	BOPAM-INFA4	L-13	BOPAM-INFA4.1	R-13
24	BOPAM-INFA5	G-3	BOPAM-INFA5.1	T-3
25	BOPAM-INFA6	L-13	BOPAM-INFA6.1	R-13
26	BOPAM-INFA7	L-13	BOPAM-INFA7.1	R-13
27	BOPAM-INFA8	G-3	BOPAM-INFA8.1	T-3
28	BOPAM-INFB1	H-5	BOPAM-INFB1.1	R-5
29	BOPAM-INFB2	H-5	BOPAM-INFB2.1	R-5
30	BOPAM-INFB3	C-12	BOPAM-INFB3.1	S-12
31	BOPAM-INFB4	L-15	BOPAM-INFB4.1	F-15
32	BOPAM-INFB5	L-15	BOPAM-INFB5.1	M-15
33	BOPAM-INFB6	L-8	BOPAM-INFB6.1	F-8
34	BOPAM-RSV1	S-3	BOPAM-RSV1.1	N-3
35	BOPAM-RSV2	S-3	BOPAM-RSV2.1	N-3
36	BOPAM-RSV3	V-3	BOPAM-RSV3.1	N-3
37	BOPAM-RSV4	T-3	BOPAM-RSV4.1	N-3
38	BOPAM-RSV5	T-13	BOPAM-RSV5.1	R-13
39	BOPAM-RSV6	S-3	BOPAM-RSV6.1	N-3
40	BOPAM-RSV7	S-3	BOPAM-RSV7.1	N-3
41	BOPAM-RSV8	Y-3	BOPAM-RSV8.1	N-3
42	BOPAM-RSV9	I-13	BOPAM-RSV9.1	R-13
43	BOPAM-RSV10	I-13	BOPAM-RSV10.1	R-13
44	BOPAM-RSV11	T-13	BOPAM-RSV11.1	R-13
45	BOPAM-RSV12	I-13	BOPAM-RSV12.1	R-13
46	BOPAM-RSV13	I-13	BOPAM-RSV13.1	R-13

BOPAM is the given name of the AMPs, with the following two letters being derivatives from the pathogens, for example, *Klebsiella pneumoniae* = KP. The numbers without periods represent the parental AMPs of the pathogens, while those with periods represent their derivatives.

**Table 3 bioengineering-09-00305-t003:** Physicochemical properties of the mutated antibacterial and antiviral pneumonia AMPs.

S/N	AMPs	Molecular Mass (Da)	% Hydrophobic	Common Amino Acid	Net Charge	PI	Boman Index (kcal/mol)	Half-Life in Mammals (Hours)
1	BOPAM-AB1.1	1783.33	52	L	+2	11.65	−1	1.1
2	BOPAM-AB2.1	1817.249	52	L	+2	11.65	−0.88	1.1
3	BOPAM-AB3.1	1937.58	64	L	+2	11.65	−1.63	1.1
4	BOPAM-AB4.1	1867.50	58	L	+2	10.81	−1.66	1.1
5	BOPAM-AB5.1	1714.1	47	G	+4	11.92	0.72	4.4
6	BOPAM-KP1.1	2496.90	30	SC	0	7.12	2.26	100
7	BOPAM-KP2.1	2297.701	38	N	+2	8.79	1.32	30
8	BOPAM-KP3.1	2406.25	42	N	+2	8.82	1.61	30
9	BOPAM-KP4.1	2589.19	28	K	+4	10.58	1.86	20
10	BOPAM-KP5.1	2694.32	39	K	+4	9.66	2.44	20
11	BOPAM-KP6.1	2357.05	38	K	+8	10.98	2.64	5.5
12	BOPAM-KP7.1	2600.13	39	S	+2	8.83	2.35	5.5
13	BOPAM-SP1.1	5084.34	27	R	+4	9.49	3.67	0.8
14	BOPAM-SP2.1	4904.86	25	G	+4	9,47	3.11	1.3
15	BOPAM-SP3.1	4793.03	45	G	−3	5.43	0.59	1.1
16	BOPAM-SP4.1	4807.06	45	G	−3	5.43	0.57	1.1
17	BOPAM-SP5.1	4807.06	45	G	−3	5.44	0.55	1.1
18	BOPAM-SP6.1	4892.21	47	V	−1	6.78	0.72	2.8
19	BOPAM-SP7.1	5168.24	25	K	+5	9.85	3.48	1.3
20	BOPAM-NFA1.1	1569.43	28	P	0	6.34	1.57	7.2
21	BOPAM-NFA2.1	1555.28	57	VIL	+1	8.55	0.52	1.2
22	BOPAM-NFA3.1	1562.33	28	T	−2	3.49	1.63	7.2
23	BOPAM-NFA4.1	1587.47	28	P	0	6.34	1.75	7.2
24	BOPAM-NFA5.1	1585.69	28	TOP	−1	3.75	1.15	7.2
25	BOPAM-NFA6.1	1596.50	28	P	+1	9.69	1.78	7.2
26	BOPAM-NFA7.1	1605.29	50	ISV	+1	8.55	1.03	1.2
27	BOPAM-NFA8.1	1586.40	28	PT	−2	3.55	1.24	7.2
28	BOPAM-NFB1.1	1888.60	40	R	+2	12.20	2.95	30
29	BOPAM-NFB2.1	1863.61	46	MRVP	+1	10.40	1.8	30
30	BOPAM-NFB3.1	1671.56	40	NL	0	5.84	1.25	5.5
31	BOPAM-NFB4.1	1798.04	66	L	+2	10.81	−1.44	5.5
32	BOPAM-NFB5.1	1786.94	66	L	+1	9.70	−1.68	5.5
33	BOPAM-NFB6.1	1690.34	46	N	0	5.76	0.7	5.5
34	BOPAM-RSV1.1	1892.35	43	IKN	+1	8.54	1.6	20
35	BOPAM-RSV2.1	1803.10	31	N	−1	4.43	2.49	1
36	BOPAM-RSV3.1	1744.03	37	N	0	6.45	2.36	1.4
37	BOPAM-RSV4.1	1797.11	43	N	−1	4.18	1.95	1.3
38	BOPAM-RSV5.1	1742.30	37	N	+2	11.65	2.77	4.4
39	BOPAM-RSV6.1	1840.79	50	I	−1	4.18	1.09	1.3
40	BOPAM-RSV7.1	1854.40	31	N	0	6.41	3.01	1.4
41	BOPAM-RSV8.1	1734.98	43	N	−1	3.85	1.41	1.9
42	BOPAM-RSV9.1	1816.39	25	N	+2	11.65	3.27	1.4
43	BOPAM-RSV10.1	1815.40	31	N	+1	9.69	2.99	100
44	BOPAM-RSV11.1	1729.96	37	N	−1	4.11	2.39	1.4
45	BOPAM-RSV12.1	1847.16	37	N	0	6.45	2.48	1.1
46	BOPAM-RSV13.1	1985.35	25	DR	+1	9.53	5.36	1.3

BOPAM is the given name of the AMPs, with the following two letters being derivatives from the pathogens, for example, *Klebsiella pneumoniae* = KP. The numbers with periods represent the derivatives. BOPAM-AB = anti-*Acinetobacter baumanni* AMPs; BOPAM-INFA = anti-*influenza A* virus AMPs; BOPAM-INFB = anti-*influenza B* virus AMPs; BOPAM-RSV = anti-*respiratory syncytial* virus AMPs; BOPAM-SP = anti-*Streptocococcus pneumoniae* AMPs; BOPAM-KP = anti-*Klebsiella pneumoniae* AMPs.

**Table 4 bioengineering-09-00305-t004:** Structure prediction scores of the anti-pneumonia AMPs.

S/N	AMP Name	C-Score	TM-Score	RMSD
1	BOPAM-AB1.1	−0.45	0.66 ± 0.13	1.5 ± 1.4 Å
2	BOPAM-AB2.1	−0.44	0.66 ± 0.13	1.5 ± 1.4 Å
3	BOPAM-AB3.1	−0.16	0.69 ± 0.12	1.0 ± 1.0 Å
4	BOPAM-AB4.1	−0.33	0.67 ± 0.13	1.3 ± 1.3 Å
5	BOPAM-AB5.1	0.42	0.77 ± 0.10	0.5 ± 0.5 Å
6	BOPAM-KP1.1	0.71	0.81 ± 0.09	0.5 ± 0.5 Å
7	BOPAM-KP2.1	−1.45	0.54 ± 0.15	3.7 ± 2.6 Å
8	BOPAM-KP3.1	−1.42	0.54 ± 0.15	3.7 ± 2.5 Å
9	BOPAM-KP4.1	−1.56	0.52 ± 0.15	3.9 ± 2.7 Å
10	BOPAM-KP5.1	0.03	0.72 ± 0.11	1.2 ± 1.2 Å
11	BOPAM-KP6.1	−0.02	0.71 ± 0.12	1.2 ± 1.2 Å
12	BOPAM-KP7.1	−0.57	0.64 ± 0.13	2.3 ± 1.8 Å
13	BOPAM-SP1.1	0.04	0.72 ± 0.11	2.3 ± 1.8 Å
14	BOPAM-SP2.1	−0.12	0.70 ± 0.12	2.6 ± 1.9 Å
15	BOPAM-SP3.1	−1.88	0.49 ± 0.15	6.1 ± 3.8 Å
16	BOPAM-SP4.1	−1.88	0.49 ± 0.15	6.1 ± 3.8 Å
17	BOPAM-SP5.1	−1.97	0.48 ± 0.15	6.3 ± 3.8 Å
18	BOPAM-SP6.1	−1.47	0.53 ± 0.15	5.2 ± 3.4 Å
19	BOPAM-SP7.1	−0.14	0.70 ± 0.12	2.7 ± 2.0 Å
20	BOPAM-INFA1.1	−1.03	0.58 ± 0.14	2.2 ± 1.7 Å
21	BOPAM-INFA2.1	−0.14	0.70 ± 0.12	0.7 ± 0.7 Å
22	BOPAM-INFA3.1	−1.50	0.53 ± 0.15	3.1 ± 2.2 Å
23	BOPAM-INFA4.1	−1.05	0.58 ± 0.14	2.2 ± 1.7 Å
24	BOPAM-INFA5.1	−0.97	0.59 ± 0.14	2.1 ± 1.7 Å
25	BOPAM-INFA6.1	−1.00	0.59 ± 0.14	2.1 ± 1.7 Å
26	BOPAM-INFA7.1	−0.71	0.62 ± 0.14	1.6 ± 1.4 Å
27	BOPAM-INFA8.1	−1.13	0.57 ± 0.14	2.4 ± 1.8 Å
28	BOPAM-INFB1.1	−0.77	0.62 ± 0.14	1.9 ± 1.5 Å
29	BOPAM-INFB2.1	−0.78	0.61 ± 0.14	1.9 ± 1.6 Å
30	BOPAM-INFB3.1	−1.19	0.57 ± 0.15	2.6 ± 1.9 Å
31	BOPAM-INFB4.1	−0.03	0.71 ± 0.12	0.6 ± 0.6 Å
32	BOPAM-INFB5.1	−0.35	0.67 ± 0.13	1.1 ± 1.1 Å
33	BOPAM-INFB6.1	−0.91	0.60 ± 0.14	2.1 ± 1.7 Å
34	BOPAM-RSV1.1	−0.00	0.71 ± 0.11	0.7 ± 0.7 Å
35	BOPAM-RSV2.1	−0.01	0.71 ± 0.11	0.7 ± 0.7 Å
36	BOPAM-RSV3.1	−0.23	0.68 ± 0.12	1.1 ± 1.1 Å
37	BOPAM-RSV4.1	−1.53	0.53 ± 0.15	3.4 ± 2.3 Å
38	BOPAM-RSV5.1	−0.37	0.67 ± 0.13	1.3 ± 1.3 Å
39	BOPAM-RSV6.1	−0.85	0.61 ± 0.14	2.1 ± 1.7 Å
40	BOPAM-RSV7.1	−0.13	0.70 ± 0.12	0.9 ± 0.9 Å
41	BOPAM-RSV8.1	−1.42	0.54 ± 0.15	3.2 ± 2.2 Å
42	BOPAM-RSV9.1	−0.11	0.70 ± 0.12	0.9 ± 0.9 Å
43	BOPAM-RSV10.1	−0.78	0.61 ± 0.14	2.0 ± 1.6 Å
44	BOPAM-RSV11.1	−1.26	0.56 ± 0.15	2.9 ± 2.1 Å
45	BOPAM-RSV12.1	−1.46	0.53 ± 0.15	3.2 ± 2.3 Å
46	BOPAM-RSV13.1	−0.10	0.70 ± 0.12	0.8 ± 0.8 Å

BOPAM-AB = anti-*Acinetobacter baumanni* AMPs; BOPAM-INFA = anti-*influenza A* virus AMPs; BOPAM-INFB = anti-*influenza B* virus AMPs; BOPAM-RSV = anti-*respiratory syncytial* virus AMPs; BOPAM-SP = anti-*Streptocococcus pneumoniae* AMPs; BOPAM-KP = anti-*Klebsiella pneumoniae* AMPs.

**Table 5 bioengineering-09-00305-t005:** Docking analysis only of derivative viral and bacterial anti-pneumonia AMPs and their receptors which showed increased binding scores.

S/N	Receptors	Parental AMPs	Binding Score	Mutated AMPs	Binding Score
1	Iron-Regulated OMP	BOPAMAB1	10,566	BOPAMAB1.1	11,826
2	Iron-Regulated OMP	BOPAMAB5	11,388	BOPAMAB5.1	12,952
3	Nucleoprotein	BOPAMINFA1	12,134	BOPAMINFA1.1	12,346
4	Nucleoprotein	BOPAMINFA2	10,870	BOPAMINFA2.1	10,888
5	Nucleoprotein	BOPAMINFA3	10,572	BOPAMINFA3.1	12,578
6	Nucleoprotein	BOPAMINFA4	12,300	BOPAMINFA4.1	12,458
7	Nucleoprotein	BOPAMINFA6	12,110	BOPAMINFA6.1	13,256
8	Nucleoprotein	BOPAMINFA7	10,982	BOPAMINFA7.1	11,120
9	Nucleoprotein	BOPAMINFA8	12,604	BOPAMINFA8.1	14,170
10	Nucleoprotein	BOPAMINFB5	11,704	BOPAMINFB5.1	11,932
11	Iron-Regulated Outer Membrane Protein	BOPAMKP1	11,305	BOPAMKP1.1	12,268
12	Iron-Regulated Outer Membrane Protein	BOPAMKP3	12,384	BOPAMKP3.1	13,216
13	Iron-Regulated Outer Membrane Protein	BOPAMKP5	10,810	BOPAMKP5.1	10,984
14	Iron-Regulated Outer Membrane Protein	BOPAMKP6	13,208	BOPAMKP6.1	13,870
15	Chain A Protein	BOPAMRSV3	9068	BOPAMRSV3.1	9156
16	Chain A Protein	BOPAMRSV6	8194	BOPAMRSV6.1	9236
17	Chain A Protein	BOPAMRSV9	8500	BOPAMRSV9.1	9278
18	Chain A Protein	BOPAMRSV10	9020	BOPAMRSV10.1	9158
19	Chain A Protein	BOPAMRSV13	8866	BOPAMRSV13.1	9072
20	Pneumolysin	BOPAMSP1	12,306	BOPAMSP1.1	13,164
21	Pneumolysin	BOPAMSP3	12,116	BOPAMSP3.1	14,134
22	Pneumolysin	BOPAMSP4	12,384	BOPAMSP4.1	12,934
23	Pneumolysin	BOPAMSP7	11,830	BOPAMSP7.1	12,378

BOPAM is the given name of the AMPs, with the following two letters being derivatives from the pathogens, for example, *Klebsiella pneumoniae* = KP. The number without periods represents the parental AMPs of the pathogens, while those with periods represent their derivatives. BOPAM-AB = anti-*Acinetobacter baumanni* AMPs; BOPAM-INFA = anti-*influenza A* virus AMPs; BOPAM-INFB = anti-*influenza B* virus AMPs; BOPAM-RSV = anti-*respiratory syncytial* virus AMPs; BOPAM-SP = anti-*Streptococcus pneumoniae* AMPs; BOPAM-KP = anti-*Klebsiella pneumoniae* AMPs.

## Data Availability

The data used for this work were generated in our previous studies which are referenced in the Introduction section. The parental antimicrobial peptides used for this work are provided in the Appendix A section along with the databases’ accession numbers.

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
