# Peer review of "Analytical Studies of Antimicrobial Peptides as Diagnostic Biomarkers for the Detection of Bacterial and Viral Pneumonia"

_bioengineering, 2022, doi:10.3390/bioengineering9070305_

Round 1

Reviewer 1 Report

In this study, the authors used bioinformatics tools to predict the derivates of antimicrobial peptides (AMPs) as potential diagnostic biomarkers for the detection of bacterial and viral pneumonia to distinct from other associated diseases, such as atherosclerosis plaques. The design of the study is fine. There are some comments for the results. Here are the specific suggestions for the revision:

Abbreviations, defining HIV in the abstract and context, SDM is listed in line 58, then other parts, such as line 61, 68, etc, should use the abbreviate, and checking all the others across the manuscript.

In lines 47-52, a significant role of AMPs is missing, immune regulation (e.g., PMID: 35326812, PMID: 32457759).

The font of text changed across the manuscript.

In table 1, some part of the data was not shown in the manuscript. Like the Note under Table 3, Tables 1, and 2, should also include the meaning of BOPAM-AB and others.

In figure 1, BOPAM-SP7.1 was separated.

Figure 2, please explain why the binding affinity was increased from these derivatives (Line 281). The ribbon diagram of the receptor does not show the peptide-binding site, using the surface may show a clearer binding site or adding the sticks to the binding site.

Author Response

Dear Reviewer,

I trust this email finds you well.

Please see the response to the comments as attached.

Reviewer 2 Report

Line 23 and later in the text  (115, 128, 157, 192, 257, etc):  pneumonia receptors

Comments: pneumonia receptores do not exist. Some bacterial or viral proteins perhaps can bind with AMP. And some of these bacteria and viruses may course pneumonia.

Line 25: pneumonia proteins

Comments: pneumonia proteins do not exist. You shuold speak about proteins from bacteria and viruses, which course pneumonia.

Lines 26, 156, 158, 198, 202, 210, 212, 259:  anti-pneumonia properties

Comments: if some peptides shwoed in silico the ability to bind with some bacterial proteins it does not mean that they will have antibacterial activity.You should not use this words, it is not proven, that this AMPs  eliminate bacteria or viruses, that course pneumonia 

Lines 55-57:  A long-lasting AMP is desirable to overcome these previously mentioned constraints for applications in biotechnology and medical sciences as potential diagnostic biomarkers.

Comments: for diagnostic purpose does not matter short half-lives and severe proteolytic activities of AMPs

Lines 92-96: If you suggest the usage of AMPs for testing bacterial and viral agents, it is necessary to discribe the existig tests on the base of peptides.

Line 127: 2.5. Identification of Receptors

Comments: Identification of receptors may be proven only by laboratory experiments. It is better to use Prediction of Receptors 

Lines 264-302, figure 2:

Comments: If you are speaking about outer membrane  proteins, interacting with AMPs, you should analyse the binding of AMPs only with extracellular fragment of the protein. 

In Conclusions: What is the advantage of the AMPs calculated in this article in comparison with the AMPs described in the publication?

In the article Bakare et al. BMC Molecular and Cell Biology, 2020 

Author Response

Dear Reviewer,

Thank you for your feedback.

I hereby upload a response to the comments as attached.

Reviewer 3 Report

In this manuscript, the authors aimed to use the parental anti-pneumonia AMPs generated from the authors’ previous studies, using in silico binding studies as templates to create derivative AMPs with greater affinity for detecting bacterial and viral pneumonia.

The parental antimicrobial peptides were discovered in the authors’ previous research in silico analyses, published in 2020 and 2021 (BMC Molecular and Cell Biology & Scientific reports). In this research, based on the parental antimicrobial peptides, the authors generated derivative AMPs with greater affinity to detect bacterial and viral pneumonia using the in silico site-directed mutagenesis approach and emphasized the probability of AMPs as diagnostic biomarkers in the detection of bacterial and viral pneumonia.

Major comment:

1.        For in silico site-directed mutagenesis of the AMPs, “All amino acids substituted to generate derivate AMPs were of similar characteristics to the amino acids present in the parental AMPs to maintain the predicted activity and functioning of the AMPs.” was indicated on line 158-160. However, in the mutated AMPs, for examples: BOPAM-SP1 to BOPAM-SP7, the uncharged amino acids, asparagine, serine and leucine, were substituted by the charged amino acids, aspartic acid and arginine, and some of the other mutated AMPs as well. Could the authors explain more in detail about what characteristics of amino acids were referenced in the mutated AMPs operation?

2.        Why were the parental AMPs not analyzed in Table 3? These data should be supplied. The authors should also discuss the differences in physicochemical properties between parental antimicrobial peptide sequence and the derived antimicrobial peptide sequence.

3.        The graphic abstract or the schematic illustration need to be provided for presentation on how to use anti-pneumonia AMPs as templates to identify derivative AMPs in the diagnosis of pneumonia by lateral flow device.

Minor comment:

1.       Please check the font size of the whole manuscript.

2.       Please reconfigure the position of tables and pictures to the center of the article.

3.        In Section 2.1, the number of antibacterial peptides should be checked in the manuscript. The sum of AMPs against bacteria was 18 peptides rather than 19.

4.        In Section 2.1, antibacterial AMPs should be rewritten as ‘antibacterial peptides’ or ‘AMPs’.

5.       In the legend of Figure 1, please confirm and revise the word “3-D structures…”

6.       In the legend of Figure 1, please uniform the noun “anti pneumonia” or “anti-pneumonia” in the article.

7.       Please supply the whole sequence of the parental AMPs and derivative AMPs in the article or in the supplementary data.

8.        In Table 3, “Half-life in mammals (Hrs)” was shown in the physicochemical properties of the derivative AMPs’ analysis. Could the authors illustrate which parameters were calculated in APD3 and BACTIBASE servers for this data?

9.        In Table 4, the deviation range of RMSD was very wide. However, in the content, “The RMSD values were above 1Å further strengthening that the AMPs were correctly predicted.” was indicated in line 250. Are there any interpretations for the data?

Author Response

(The authors gave the same response as above.)

Round 2

Reviewer 1 Report

Authors revised the manuscript accordingly, no additional comments. 

Author Response

Many thanks for your feedback.

Reviewer 2 Report

Line 54: When you are speaking about immunomodulatory function it is necessary to cite the article:

 Guryanova, S.V.; Ovchinnikova, T.V. Immunomodulatory and Allergenic Properties of Antimicrobial Peptides. Int. J. Mol. Sci. 2022, 23, 2499. https://doi.org/10.3390/ijms23052499

Author Response

Many thanks for your feedback.

We hereby address the comments as follows.

Reviewer 3 Report

The authors have answered my questions.

Author Response

Many thanks for your feedback.